# Multivariate Time Series Forecasting under Hyperbolic Space Hierarchical Constraints

## Abstract

Multivariate time series forecasting has experienced a surge in interest recently. However, significant challenges remain in effectively modeling the multi-level dependencies among time points, sequences, and channels. Existing methods often struggle to fully capture the hierarchical relationships between these three aspects or face efficiency issues. To address this, we propose **HyperTime**: **Hyper**bolic space hierarchical constraints for multivariate **Time** series forecasting. This method initially segments the time series into patches and then extracts temporal dependencies to obtain representations for each channel. It subsequently derives interrelationships among multiple channels based on these representations, encoding time patches, individual channels, and multi-channel series into a unified hyperbolic representation space. By imposing hyperbolic hierarchy and entailment constraints on the encoded representations, the method leverages relationships from local to global among the three levels, ensuring sufficient interactions among point, intra- and inter-channel information. We evaluated HyperTime on several commonly used multivariate time series forecasting datasets and compared it with previously top-performing models. The experimental results demonstrate the effectiveness and efficiency of HyperTime, achieving state-of-the-art performance with only linear complexity. This highlights its proficiency in capturing complex temporal dependencies and interrelationships among channels. Our code is included in the supplemental material and will be released open-source.

## 1 Introduction

The multivariate time series, characterized by its chronological sequences and multi-channel variables, plays a crucial role in various fields such as energy management, weather prediction, and traffic flow. Consider the specific hierarchical structure: within a single channel (one variable), contiguous segments of time (time patches) form a natural hierarchy where finer-grained time patches are subsumed by coarser ones. This reflects the idea that short-term fluctuations aggregate to form longer-term trends or patterns Rangapuram et al. (2023). Furthermore, across multiple channels (different variables), the individual channels can be seen as components contributing to a higher-level system. Thus, a single channel represents a lower level in the hierarchy relative to the collective behavior of all channels Naghashi et al. (2025). A key aspect of forecasting often lies in uncovering the inherent hierarchical relationships among time patches, single channel and multiple channels.

In recent years, the field of multivariate time series forecasting has witnessed an upsurge. Numerous studies have demonstrated that deep learning methods significantly outperform traditional approaches, establishing deep learning forecasters as the current forefront of research in multivariate time series forecasting. The MLP-based models, such as RLinear Li et al. (2023b), expand upon reversible instance normalization Kim et al. (2022) and linear regression to manage intricate temporal dependencies. Models like N-BEATS Oreshkin et al. (2020), TiDE Das et al. (2023), and DLinear Zeng et al. (2023) dissect and forecast the fundamental components of time series data, including trends and seasonality. N-HiTS Challu et al. (2023), TimeMixer Wang et al. (2024a), and WP-Mixer Murad et al. (2025) process input data hierarchically across multiple scales. SOFTS Han et al. (2024) utilizes inverse embedding and a star aggregate-redistribute module to effectively model the interrelationships among multiple channels. As for CNN-based models, SCINet Liu et al. (2022a) harnesses temporal attention mechanisms to adeptly manage long-term dependencies. TimesNet Wu et al. (2023) integrates sparsity and explainability to enhance both prediction accuracy. Turning to

RNN-based models, MLST Lai et al. (2018) employs sparse tensor decomposition to capture complex temporal dependencies. DeepAR Salinas et al. (2020) employs a likelihood function to predict values and their associated uncertainties. Transformer-based models have made significant strides. Autoformer Chen et al. (2021) utilizes auto-covariance, while FEDformer Zhou et al. (2022) integrates frequency-domain decomposition to handle long-term dependencies. PatchTST Nie et al. (2023) employs a patch-based approach to effectively handle long sequences, and iTransformer Liu et al. (2024) proposes inverse embedding for efficient multivariate modeling. TimeXer Wang et al. (2024b), which benefits from both PatchTST and iTransformer, has achieved promising results. The GNN-based methods Wu et al. (2020); Liu et al. (2022b), such as MSGNet Cai et al. (2024), effectively capturing both complex temporal and spatial dependencies inherent in multivariate data.

However, existing forecasters primarily face the following challenges: Channel-independent Han et al. (2023) methods fail to model the hierarchy between single channel and multiple channels, limiting their performance on scenarios with multiple variables. Channel-dependent methods either do not adequately model the hierarchy between time points and single channel to maintain high efficiency Liu et al. (2024), or suffer from poor computational efficiency when simultaneously processing the three-level hierarchy Wang et al. (2024b); Cai et al. (2024). Furthermore, all of these methods model the inherent hierarchical relationships in Euclidean space, where the distance between two points grows linearly with their separation, which is insufficient for representing the multi-level, tree-like structures often found in hierarchical data Li et al. (2023a). Additionally, Euclidean space requires high-dimensional representations to adequately capture complex hierarchical relationships, which increases computational costs and risks overfitting. In contrast, hyperbolic space, with its negative curvature, allows distances to grow exponentially as they approach the boundary, making it a more natural fit for hierarchical modeling. These limitations highlight the need for alternative geometric frameworks, such as hyperbolic space, which better align with the intrinsic exponential nature of hierarchical relationships Pan & Wang (2021).

Therefore, this paper proposes a novel approach, HyperTime, to investigate the application of hyperbolic space learning to multivariate time series data, specifically aiming to capture the hierarchy between time patch, single channel and multiple channels. We hypothesize that by leveraging the intrinsic properties of hyperbolic geometry, we can uncover and represent these hierarchical relationships more effectively and efficiently than Euclidean methods, leading to improved performance in downstream tasks that benefit from an understanding of the data's hierarchical organization. The contributions can be summarized as follows:

- We propose HyperTime, a simple and novel framework to simultaneously aggregate the three-level hierarchy between time patch, single channel and multiple channels, in a unified hyperbolic representation space, with only linear computational complexity.

- HyperTime innovatively and effectively introduces Hyperbolic Space Hierarchical Constraints into multivariate time series forecasting, demonstrating the importance of hyperbolic representations in learning the hierarchy and entailment of relationships.

- When compared to previous top-performing methods, HyperTime exhibits exceptional proficiency in multivariate time series forecasting, achieving state-of-the-art results in most scenarios, especially for large-scale and non-stationary data.

## 2 RELATED WORK

### 2.1 HYPERBOLIC REPRESENTATION LEARNING

Hyperbolic representation learning has garnered significant attention in various fields of deep learning, offering a novel approach to capturing complex hierarchical relationships within data. Unlike traditional Euclidean space, hyperbolic space is characterized by a constant negative curvature, which allows it to naturally represent hierarchical structures. For example, hyperbolic neural networks Ganea et al. (2018b) and hyperbolic graph neural networks Liu et al. (2019) leverage the unique properties of hyperbolic geometry to improve the representation of hierarchical relationships, leading to more accurate and efficient learning. Additionally, hierarchical semantics in language Everaert et al. (2015) have been leveraged to embed textual data in hyperbolic space Dhingra et al. (2018). In the vision domain, research has explored the integration of hyperbolic embeddings

with other deep learning paradigms, such as contrastive learning Pal et al. (2024), entailment learning Pal et al. (2024); Ganea et al. (2018a); Desai et al. (2023), and generative models, to improve the robustness and generalization of learned representations. Despite these promising results, hyperbolic representation learning still faces challenges, such as the computational complexity associated with hyperbolic operations and the need for further theoretical understanding of hyperbolic space in machine learning. Ongoing research efforts are dedicated to addressing these challenges, aiming to unlock the full potential of hyperbolic geometry in advancing the field of representation learning.

## 2.2 Time Series Forecasting Models

The channel-independent forecasters separately analyze and predict each time series channel without considering the potential correlations or interactions between different channels. Examples of such methods include regression-based approaches like DeepAR Salinas et al. (2020) and RLinear Li et al. (2023b), frequency-based methods like FreTS Yi et al. (2023) and FEDformer Zhou et al. (2022), decomposition-based methods like N-BEATS Oreshkin et al. (2020), N-HiTS Challu et al. (2023), and PDF Dai et al. (2024), as well as the patch-based PatchTST Nie et al. (2023). The primary advantage of channel-independent forecasting is its simplicity and high efficiency, making it suitable for situations where the channels are unrelated or when computational resources are limited. However, these approaches may not be optimal when there are significant interrelationships among the channels, as they fail to leverage the full information of the multivariate data.

In contrast, channel-dependent forecasters involve predicting future values of multiple time series by considering the interrelationships among the channels. For example, TimesNet Wu et al. (2023) utilizes convolutional layers to simultaneously model intra- and inter-channel dependencies. SOFTS Han et al. (2024) and iTransformer Liu et al. (2024) embed each channel into a single vector using linear transformations, then extract inter-channel dependencies through the encoder. TimeXer Wang et al. (2024b) combines the advantages of PatchTST and inter-channel dependency modeling, while TimeMixer++ Wang et al. (2025) incorporates multi-scale decomposition, frequency analysis, and inter-channel information. MSGNet Cai et al. (2024) learns multi-scale inter-channel correlations through GNN blocks. These models are able to achieve promising results across various scenarios. However, due to the complex computations for each channel, their low efficiency becomes unacceptable for large-scale and multi-channel data. Therefore, it urgently requires the ability to model intra- and inter-channel dependencies simultaneously and efficiently.

## 3 Methodology

### 3.1 Preliminary

Hyperbolic space, a non-Euclidean geometry with constant negative curvature, $c < 0$, which leads the exponential growth of circumference with radius and the divergence of geodesics. This unique property is distinct from the flat Euclidean space, making it an ideal choice for learning representations of data with an inherent hierarchical structure Pal et al. (2024). Among its mathematical representations, the Lorentz model stands out for its analytical tractability and geometric intuition Nickel & Kiela (2018), where hyperbolic space $\mathbb{H}^n$ is realized as the upper sheet of a two-sheeted hyperboloid embedded in a Minkowski space $\mathbb{R}^{n+1}$, equipped with the inner product $\langle X, Y \rangle = -x_0 y_0 + \sum_{i=1}^{n} x_i y_i$, where $x_0 = \sqrt{\|x\|^2 - 1/c}$. Points in $\mathbb{H}^n$ satisfy $\langle X, X \rangle = 1/c$ and $x_0 > 0$, with geodesics corresponding to intersections of the hyperboloid with planes through the origin. The distance and angle can be computed as $\mathfrak{D}_{xy} = \sqrt{-1/c} \cdot cosh^{-1}(c\langle X, Y \rangle)$ and $\mathfrak{A}_{xy} = cos^{-1}((y_0 - x_0 c\langle X, Y \rangle)/(\|x\|\sqrt{(c\langle X, Y \rangle)^2 - 1}))$.

Besides, we provide a formal description of the time series forecasting task. The time series data is defined as $X = \{x_1, x_2, \ldots, x_t\}, x \in R^d$, where $t$ is the current time and $d$ is the dimension of features of each time point data. The goal is to predict the sequence $Y = \{x_{t+1}, x_{t+2}, \ldots, x_{t+h}\}$, where $h$ is the prediction horizon. We propose a novel method, HyperTime, which mainly includes three components: patch embedding, HyperTime encoding, and prediction decoding. The overall architecture is shown in Figure 1.

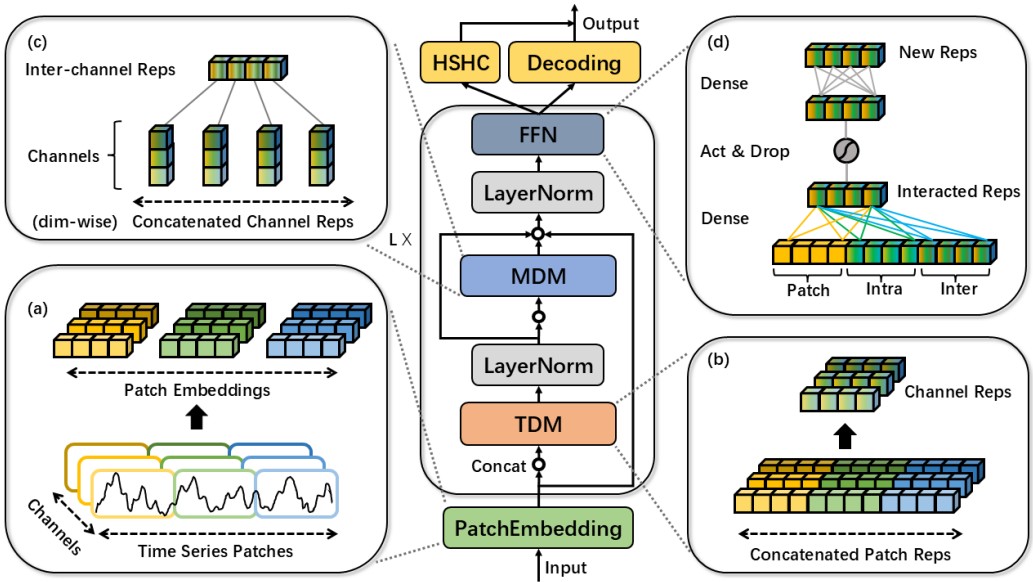

Figure 1: Overall architecture of HyperTime, includes patch embedding, HyperTime encoding, and decoding. (a) Patch embedding to embed the time series patches as tokens. (b) Temporal Dependency Modeling (TDM) from the concatenated patch representations. (c) Multi-channel Dependency Modeling (MDM) from the dimension-wise concatenated intra-channel representations. (d) Hierarchically representation aggregation. Note that the Hyperbolic Space Hierarchical Constraints (HSHC) will be introduced with details in the following of this section.

## 3.2 HYPERTIME ARCHITECTURE

**Patch Embedding.** To reduce memory usage and improve computational efficiency, a sliding window of length $k$ is defined as $T_i = \{x_{i+1}, x_{i+2}, \ldots, x_{i+k}\}$, with a step size of $s$ to divide the time series into multiple smaller patches. A fully connected linear layer is employed for patch embedding, accepting each patch within the window as input and outputting a single aggregated representation.

$$S^1 = Patch\_Embed(T) \tag{1}$$

In Equation (1), $n$ represents the number of patches, $T = \{T_1, T_2, \ldots T_n\}$. The linear operation treats multivariate time series as multiple univariate series (channel independent) Han et al. (2023), and multiplies $k$ values within the sliding window by a matrix with dimensions $k \times d$, where $d$ is the dimension of embedding. This processing method is consistent with that in PatchTST Nie et al. (2023). The method of dividing sequence data into patches for representation learning has been demonstrated in numerous related studies to have advantages in time series forecasting.

**HyperTime Encoding.** The patch data representations extracted by the input module are fed into the HyperTime encoding layer, where intra- and inter-channel information is extracted and unified within the same hyperbolic representation space for hierarchical multi-level aggregation.

$$S^{j+1} = HyperTime(S^j), j = 1, 2, \ldots, l \tag{2}$$

In Equation (2), $HyperTime$ denotes only encoding layer here, where $l$ is number of encoding layers. The detailed process of it will be introduced in the following of paper.

**Prediction Decoding** After passing through $l$ encoding layers, the output $S^{l+1}$ will be fed into decoding module to obtain the final prediction results. In Equation (3), $Projection$ is implemented by a simplest fully connected layer, which is consistent with many previous studies.

$$\hat{Y} = Projection(S^{l+1}) \tag{3}$$

### 3.3 HYPERTIME ENCODING

The output from patch embedding or previous encoding layer is treated as additional representations relative to each patch, and is concatenated with the original representations.

$$C^j = S_i^1 \circ S_{i+1}^2 \circ \cdots \circ S_n^j \tag{4}$$

In Equation (4), $\circ$ denotes data concatenation. The derived high-dimensional representation contains rich temporal information, which needs to be fully exploited to enhance the overall sequence modeling capability.

**Temporal Dependency Modeling.** Information extraction is performed on the high-dimensional representations obtained from concatenation phase. To reduce overall computational complexity of the model while ensuring the effectiveness of Temporal Dependency Modeling (TDM), a simple and efficient MLP module is chosen for this operation.

$$R^j = TDM(C^j) \tag{5}$$

The high-dimensional sequence representations are compressed to a lower dimensionality, reducing the number of parameters in the model, and serve as additional representation for the original input representation. Note that the $TDM$ could be implemented using any suitable method.

**Multi-channel Dependency Modeling.** Then, we hierarchically derive multi-channel information from sequential representations with similar process. Here, $C$ represents the number of channels.

$$D^j = MDM(R_1^j \circ R_2^j \circ \cdots \circ R_C^j) \tag{6}$$

In Equation (6), the extracted sequential representations are concatenated dimension-wise to integrate information from all channels, which is subsequently processed by the Multi-channel Dependency Modeling (MDM). The $MDM$ could also be implemented using any suitable method.

**Representation Fusion.** The extracted temporal and multi-channel dependencies are then concatenated with the original input representations or the output from previous encoding layer.

$$S^{j+1} = FFN(S^j \circ Repeat(R^j \circ Repeat(D^j))) \tag{7}$$

This operation is analogous to the residual connections commonly used in Transformer-based models and helps to enhance the model's information representation capability, improve training efficiency and effectiveness, and allow for the training of deeper networks.

### 3.4 HYPERBOLIC SPACE HIERARCHICAL CONSTRAINTS

The hyperbolic space hierarchical constraints module plays a crucial role in the entire model architecture, governing how the three levels of representations are hierarchically aggregated. This module employs the Lorentz model for hyperbolic transformation, as detailed in the research of MERU Desai et al. (2023) and the accompanying code [1]. We utilize this model to implement the hierarchy and entailment constraints. The overall methodology of this module is illustrated in Figure 2.

**Hierarchy Constraint.** Following the HyperTime encoding process, we obtain the final aggregated representations of patches ($S^l$), sequences ($R^l$), and channels ($D^l$). For brevity, we will refer to these as $S$, $R$, and $D$ in the remainder of this section. In accordance with the principles of hyperbolic hierarchy learning, local information should reside at higher levels in the space. Therefore, we propose a novel Hyperbolic Triangle Loss (HTL) of hierarchy constraint to learn relationships between three levels of representations.

$$HTL = max(\alpha \sum (\mathfrak{D}_{SR} - \mathfrak{D}_{SD}, \mathfrak{D}_{RD} - \mathfrak{D}_{SD}, \mathfrak{D}_{SR} + \mathfrak{D}_{RD} - \mathfrak{D}_{SD}), 0) \tag{8}$$

In Equation (8), $\mathfrak{D}_{XY}$ represents the computation of the hyperbolic distance between $\mathbb{L}(X)$ and $\mathbb{L}(Y)$, where $\mathbb{L}$ denotes the Lorentz transformation from Euclidean to hyperbolic space. The formulation and implementation of $\mathfrak{D}$ and $\mathbb{L}$ are publicly available in MERU. Through the HTL constraint, the sequence representation ($R$) is compelled to occupy the intermediate hierarchy between the other two representations. The parameter $\alpha$ serves as a scaling factor for adjusting the loss, analogous to the role of $\beta$ in Equation (9).

---

[1] https://github.com/facebookresearch/meru

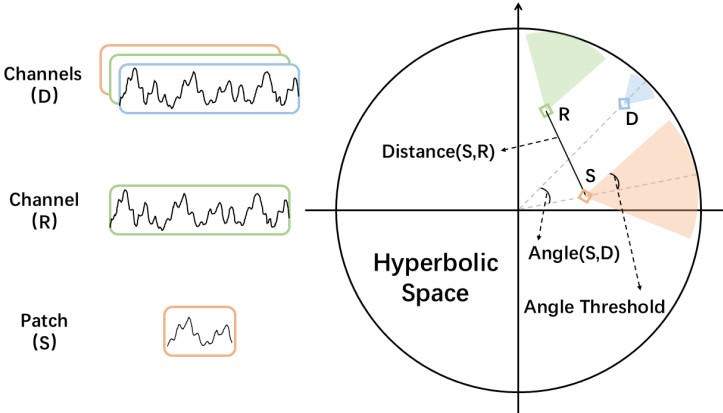

Figure 2: Hyperbolic constraints to learn the relationships of hierarchy and entailment between time patch (S), single channel (R), and multiple channels (D) in a unified hyperbolic representation space, including the illustration of distance, angle, and angle threshold. The more peripheral a point is, the lower its hierarchical level, and the more specific the information it contains.

**Entailment Constraint.** The hierarchy constraint only establishes specific levels among the three representations but does not inherently enforce entailment relationships between them. In the context of multivariate time series, patches represent local segments of sequences, while sequences themselves are local to multi-channel series. To learn these entailment relationships, we introduce a Hyperbolic Entailment Loss (HEL) for multivariate forecasting.

$$HEL = max(\beta \sum (\mathfrak{A}_{SR} - t, \mathfrak{A}_{RD} - t, \mathfrak{A}_{SD} - t), 0) \tag{9}$$

In Equation (9), $\mathfrak{A}_{XY}$ denotes the computation of hyperbolic angle between $\mathbb{L}(X)$ and $\mathbb{L}(Y)$, where $t$ is the angle threshold indicating an entailment relationship between the two.

**Constraint Learning.** During the model training phase, the predicted results are compared with the actual subsequent time series values to calculate the corresponding error. The hierarchy and entailment constraint losses are then added to this prediction error.

$$Loss = MSE(y, \hat{y}) + HTL + HEL \tag{10}$$

Model parameters are updated using the backpropagation, and MSE are utilized as error function.

## 4 EXPERIMENTS

**Datasets.** We comprehensively assessed the performance of HyperTime on eight widely used multivariate time series forecasting benchmark datasets, including ETTm1, ETTm2, ETTh1, ETTh2, Exchange, Weather, Electricity (ECL), and Traffic. These datasets, derived from real-world domains such as electricity consumption, weather records, exchange rate fluctuations, and traffic flow, are publicly available on GitHub [2]. The data processing and split ratio were consistent with those used in TimesNet Wu et al. (2023).

**Baselines.** we select a set of top-performing baselines for time series forecasting to compare their effectiveness against HyperTime. Including Transformer-based models: TimeXer Wang et al. (2024b), iTransformer Liu et al. (2024), and PatchTST Nie et al. (2023), CNN-based model: TimesNet Wu et al. (2023), MLP-based models: SOFTS Han et al. (2024) and TimeMixer Wang et al. (2024a), and GNN-based model: MSGNet Cai et al. (2024). All models were implemented using the original code provided by the authors or code meticulously replicated according to the respective papers.

**Settings.** By default, all Transformer-based models were configured with a dropout probability $p = 0.1$ and the number of attention heads $n = 16$. For patch-based methods, the patch size was set to 16 with a stride of 8, aligning with previous research. For Weather, ECL, and Traffic datasets,

---

[2]https://github.com/thuml/Time-Series-Library

both HyperTime and Transformer-based models were equipped with $l = 3$ encoding layers, and a latent dimension of $d = 512$. For smaller datasets, such as Exchange and the four ETT subsets, we employ a smaller model size to mitigate the risk of overfitting: $l = 2$ and $d = 256$. The dimension ratio for representations of patch, intra- and inter-channel representations within HyperTime was maintained at $2 : 1 : 1$ across all experimental scenarios.

Table 1: Overall experimental outcomes for multivariate time series forecasting, which are averaged from four prediction horizons: $H \in \{96, 192, 336, 720\}$ across all of the eight datasets, and the input length $L = 96$. These settings are consistent with many previous related studies. The "OOM" indicates out of memory error of TimeMixer in the corresponding scenarios.

| Model | HyperTime (Ours) | | TimeXer (2024b) | | SOFTS (2024) | | TimeMixer (2024a) | | iTransformer (2024) | | MSGNet (2024) | | PatchTST (2023) | | TimesNet (2023) | |
|---|---|---|---|---|---|---|---|---|---|---|---|---|---|---|---|---|
| Metric | MSE | MAE | MSE | MAE | MSE | MAE | MSE | MAE | MSE | MAE | MSE | MAE | MSE | MAE | MSE | MAE |
| ETTh1 | 0.438 | 0.433 | 0.450 | 0.440 | 0.457 | 0.446 | 0.470 | 0.454 | 0.448 | 0.440 | 0.497 | 0.483 | 0.448 | 0.442 | 0.523 | 0.494 |
| ETTh2 | 0.374 | 0.397 | 0.382 | 0.405 | 0.382 | 0.407 | 0.384 | 0.406 | 0.381 | 0.406 | 0.399 | 0.421 | 0.382 | 0.407 | 0.422 | 0.429 |
| ETTm1 | 0.383 | 0.396 | 0.392 | 0.402 | 0.398 | 0.404 | 0.397 | 0.400 | 0.407 | 0.410 | 0.427 | 0.438 | 0.390 | 0.401 | 0.432 | 0.430 |
| ETTm2 | 0.276 | 0.322 | 0.282 | 0.327 | 0.288 | 0.331 | 0.292 | 0.334 | 0.287 | 0.331 | 0.298 | 0.332 | 0.284 | 0.330 | 0.304 | 0.339 |
| Exchange | 0.348 | 0.395 | 0.374 | 0.411 | 0.361 | 0.405 | 0.355 | 0.399 | 0.360 | 0.405 | 0.360 | 0.406 | 0.366 | 0.407 | 0.377 | 0.416 |
| Weather | 0.251 | 0.274 | 0.247 | 0.276 | 0.257 | 0.278 | 0.259 | 0.283 | 0.260 | 0.281 | 0.254 | 0.279 | 0.255 | 0.278 | 0.262 | 0.287 |
| ECL | 0.179 | 0.274 | 0.184 | 0.280 | 0.190 | 0.276 | OOM | OOM | 0.185 | 0.275 | 0.195 | 0.298 | 0.196 | 0.283 | 0.192 | 0.294 |
| Traffic | 0.453 | 0.305 | 0.483 | 0.317 | 0.466 | 0.310 | OOM | OOM | 0.467 | 0.314 | 0.587 | 0.323 | 0.486 | 0.322 | 0.619 | 0.328 |

## 4.1 EXPERIMENTAL RESULTS

The overall results are presented in Table 1. The **bolded red** values denote the best performance on each dataset, while **bolded blue** indicate the second-best. As observed in the table, HyperTime achieves state-of-the-art results when compared with all types of recent best-performing forecasters in most scenarios, especially on non-stationary and large-scale datasets, such as Exchange and Traffic. This demonstrates the robust performance and complex dependency modeling ability of HyperTime. These results collectively demonstrate the overall effectiveness of HyperTime, and a detailed analysis of each type of method is conducted as follows.

For channel-independent methods, such as TimeMixer and PatchTST. HyperTime outperforms all by large margins across most scenarios, especially on large-scale datasets. This highlights the accurate forecasting ability of channel-dependent models. For the Exchange dataset, which comprises only 8 channels and exhibits a high degree of non-stationary, the variables have different seasonality and trends, on which non-decomposition methods may perform poorly. However, HyperTime still achieves the best, indicating its robust performance for non-stationary forecasting.

For channel-dependent methods, such as TimesNet, MSGNet, iTransformer, SOFTS, and TimeXer, these forecasters achieve much better performance compared to channel-independent methods on large datasets. However, due to the largely compromised temporal information, their prediction results for small datasets are even worse when compared with many channel-independent models. Furthermore, HyperTime surpasses them in most scenarios, especially for complex and non-stationary datasets (Traffic and Exchange), which demonstrates its robustness and effectiveness of hyperbolic space constrained hierarchical representations and the superior multivariate modeling ability.

Table 2: Ablation study with settings of original HyperTime, w/o EN (ENtailment constraint), w/o HR (HieraRchy constraint), and w/o HP (both of HyPerbolic constraints).

| Model | HyperTime | | w/o EN | | w/o HR | | w/o HP | |
|---|---|---|---|---|---|---|---|---|
| Metric | MSE | MAE | MSE | MAE | MSE | MAE | MSE | MAE |
| ETTh1 | 0.438 | 0.433 | 0.440 | 0.437 | 0.441 | 0.436 | 0.447 | 0.442 |
| Exchange | 0.348 | 0.395 | 0.353 | 0.401 | 0.351 | 0.399 | 0.358 | 0.405 |
| Weather | 0.251 | 0.274 | 0.254 | 0.280 | 0.254 | 0.279 | 0.256 | 0.281 |
| ECL | 0.179 | 0.274 | 0.183 | 0.278 | 0.185 | 0.279 | 0.189 | 0.285 |
| Traffic | 0.453 | 0.305 | 0.458 | 0.309 | 0.459 | 0.307 | 0.467 | 0.314 |

## 4.2 ABLATION STUDY

**Effectiveness of HyperTime.** Ablation study for hyperbolic constraints of hierarchy and entailment, as shown in Table 2. The results indicate that HyperTime achieves the best MSEs and MAEs in all scenarios when compared with its variants (w/o EN, HR, and HP), which demonstrate the effectiveness of hyperbolic space hierarchical constraints in learning hierarchical relationships of patch, intra- and inter-channel information, markedly improving the prediction accuracy.

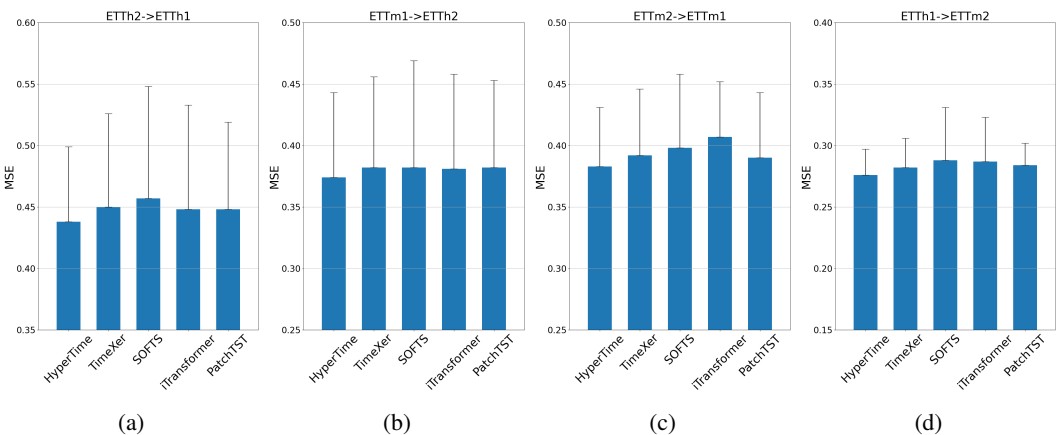

(a)          (b)          (c)          (d)

Figure 3: Zero-shot learning analysis for HyperTime. Four scenarios are adopted, which is illustrated in the figure (titles). Note that Figure (a) means the model is trained by ETTh2 and is tested by ETTh1. Each error bar in the figure denotes the MSE increment of corresponding model from original to zero-shot setting. All hyperparameters remain the same for two scenarios.

**Zero-shot Learning Analysis.** All results are presented in Figure 3. In normal settings, the performance of HyperTime is comparable to that of recent baselines. However, in zero-shot learning scenarios, HyperTime exhibits more advantages, considerably outperforming its counterparts. This superior performance strongly suggests that our model possesses robust zero-shot learning capabilities. It indicates that the model can effectively generalize to unseen data distributions without requiring explicit task-specific training.

## 4.3 LEARNING REPRESENTATIONS

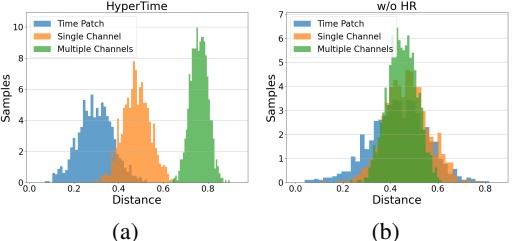

(a)          (b)

As shown in Figure 4, HyperTime successfully separates three levels of representation when compared with "w/o HR", note that there are 10% unseen from the training data. Local-levels are closer to the root, while global-levels are pushed further. This phenomenon provides an explanation for zero-shot transfer. When presented with unseen data, HyperTime can infer their relationships based on positions within hyperbolic space, which enables robust generalization beyond the scope of training data.

Figure 4: Distribution of $1K$ random selected hyperbolic representations of patch, uni- and multi-channel distances from root $R = \{\sqrt{-1/c}, 0, \ldots, 0\}, R \in \mathbb{R}^{n+1}$, with value of "bins" of histograms as 40.

## 4.4 HYPERPARAMETER SENSITIVITY

We conduct hyperparameter sensitivity analysis on dimension of latent space and number of encoding layers for HyperTime. As depicted in Figure 5, the analysis reveals that variations in both hyperparameters have a negligible impact on overall performance of HyperTime across small-scale datasets, such as ETTh1, Exchange, and Weather. For large-scale datasets like ECL and Traffic, the performance of HyperTime could benefit from increased model dimensionality and encoder layers.

The experimental results suggest that a larger amount of the latent parameters and deeper neural network should be adopted for more complex time series data.

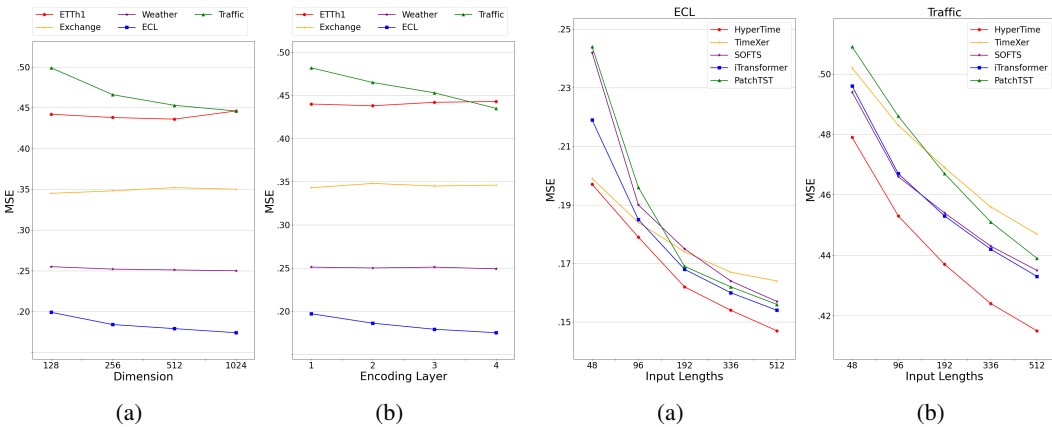

Figure 5: Hyperparameter analysis for number of dimension and encoding layers. The lower MSE indicates the better.

Figure 6: The impact of input data length on HyperTime and several strong baselines. The lower MSE indicates the better.

## 4.5 INFLUENCE OF INPUT LENGTH.

We selected two largest datasets, ECL and Traffic, to learn the impact of input data length. We chose the two because, as shown in Table 1, for small-scale datasets with simpler inter-channel dependencies, such as ETTs and Exchange, nearly all baselines could perform well. As depicted in Figure 6, HyperTime demonstrates superior effectiveness across all scenarios when compared to baselines. It achieves the best on both datasets and even surpasses others by larger margins as the length increases, which attests to the model's robust capability in modeling complex dependencies.

## 4.6 EFFICIENCY ANALYSIS

Assuming an input length $L$, a number of channels $C$, a sliding window with stride as $s$, model dimension $d$ with dim ratio $2:1:1$ of three types of representation, and prediction horizon $H$, the computational complexities are: $O(CpdL/s)$ for patch embedding, $O(Cd^2L/2s)$ for TDM, $O(CdL/2s)$ for MDM, $O(C2d^2L/s)$ for representation fusion, and $O(CHd)$ for decoding. By disregarding all constants, we derive the overall computational complexity as $O(CL + CH)$, which is linear with respect to $L$, $C$, and $H$.

Table 3: Computational complexity of encoding layers of HyperTime and typical baselines.

|            | HyperTime | TimeXer | TimeMixer | iTransformer | PatchTST |
| ---------- | --------- | --------------- | ------------- | -------------- | ---------- |
| $O(\cdot)$ | $O(CL)$   | $O(CL^2 + C^2)$ | $O(CLlog(L))$ | $O(CL + C^2)$  | $O(CL^2)$  |

## 5 CONCLUSION

In this paper, we introduce HyperTime, a novel framework that pioneers the unification of patch, univariate, and multi-channel representations within a hyperbolic representation space. By leveraging hierarchy and entailment constraints, HyperTime seamlessly integrates these heterogeneous representations with linear computational complexity. Our comprehensive evaluation demonstrates HyperTime's superior capability in multi-level information exploitation and robust zero-shot learning performance. Crucially, we establish that the effectiveness of HyperTime is intrinsically linked to the latent or explicit hierarchical structures inherent in multivariate time series data. Consequently, applying this hyperbolic geometry-based approach to domains lacking such hierarchy, such as univariate or flat-structured datasets, may result in suboptimal learning efficiency and introduce unnecessary computational overhead.

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
