# OpenReview forum: "Multivariate Time Series Forecasting under Hyperbolic Space Hierarchical Constraints"
_ICLR.cc/2026/Conference — Submitted to ICLR 2026_

### Official Review · Reviewer_QF5e · 2025-10-25

**Soundness:** 3
**Presentation:** 2
**Contribution:** 3
**Rating:** 6
**Confidence:** 4

**Summary:**

The paper introduces Hyperbolic Space Hierarchical Constraints to enhance the representation of multivariate time series. The technique includes a multi-level constraint-based loss and corresponding modules, effectively enhance the prediction performance. I think this is an interesting and insightful paper, but the current representation quality is not so well. I encourage the authors to add some more necessary experiments and improve the representation quality, which can make this paper a better work.

**Strengths:**

1. The paper provides a novel perspective for time series representation learning. Compared with E-distance constraints,  Hyperbolic Space Hierarchical Constraints can help to bring a more robust and cost-efficient representation. The overall solution, HyperTime, can bring novel insights for readers.
2. The paper provides relatively adequate experimental results, and validates the effectiveness of the proposed Hyperbolic Space Hierarchical Constraints in both ablation study and representation analysis.

**Weaknesses:**

1. Lack ablations on different hierarchical constraints. The paper only considers suming the loss of all three levels, as mentioned in Eq. 8. Do all three constraints bring positive effects to the final performance, or some may bring negative effects? Also, similar ablations should be considered in HEL(Eq. 9). Further experiments should be conducted to validate this concern.
2. Lack ablations on Hyperbolic space v.s. Euclidean space. Since the paper demonstrates the advantages (performance and effeciency) of Hyperbolic space representation constraints against Euclidean space constraint, this paper do not provide experimental results to support this. What's the model's performance when using Euclidean distance to constraint the representation? **I think this is a core experiment that supports the technical claims of the paper.**
3. Full results on all specific predicition windows should be provided, as the aggregated metrics (average) may lose much of the detailed information from the experimental results. Also, some results differ significantly from those reported in the original paper (SOFTS) and existing benchmarks (patchtst in TFB), which may affect the credibility of the findings. Could the authors provide some explanations for this discrepancy?
4. Some presentation errors and suggestions . (1) Wrong cite format. Please revise your cite format in this paper. (2) I recommend the authors to adjust some tables' size in the paper for a better representation.

**Questions:**

Please refer to the Weakness section.

---

### Official Review · Reviewer_YUyP · 2025-10-28

**Soundness:** 2
**Presentation:** 2
**Contribution:** 2
**Rating:** 2
**Confidence:** 4

**Summary:**

This paper introduces HyperTime, a framework for multivariate time series forecasting that leverages hierarchical constraints in hyperbolic space. HyperTime segments time series into patches, extracts intra- and inter-channel dependencies, and encodes these into a unified hyperbolic representation space using the Lorentz model. The main contribution lies in introducing two novel constraints—Hyperbolic Triangle Loss (hierarchy constraint) and Hyperbolic Entailment Loss (entailment constraint)—to enforce and model multi-level dependencies across time patches, single channels, and multiple channels. Extensive experiments on several standard benchmarks demonstrate HyperTime’s competitive performance and efficiency against state-of-the-art baselines, with comprehensive ablation, zero-shot, and hyperparameter sensitivity analyses supporting its claims.

**Strengths:**

**S1 Principled Motivation & Sound Architecture**: The proposal to model explicit hierarchical relationships in multivariate time series via hyperbolic geometry is conceptually sound, building on the exponential growth property of hyperbolic space to mirror hierarchical structures (see Figure 2). The formulation of hierarchy and entailment constraints distinguishes local-to-global semantics in the learned representations.

**S2 Efficiency and Scalability Demonstration**: The computational complexity is carefully analyzed (see Table 3), showing linear scaling in both the number of channels and input length. This positions the method as more efficient than many existing alternatives without sacrificing performance, as further reflected in Figure 6.

**Weaknesses:**

**W1 Limited Hyperbolic Representation Justification and Analysis:** While Section 3 provides some theoretical rationale for hyperbolic modeling, it largely relies on analogies or references to existing works (e.g., “hyperbolic allows distances to grow exponentially” on Page 2) rather than offering civil, dataset-specific evidence that such explicit hierarchy structures exist widely in the evaluated benchmarks. The approach, motivated by presumed hierarchy, would benefit greatly from a more directly empirical or formal characterization (e.g., quantifying or visualizing the presence of such hierarchies in standard datasets before and after applying hyperbolic constraints).
Figure 2 offers schematic illustration only; a more formal theoretical or empirical analysis justifying the claim that patches, single channels, and multi-channels indeed follow a strict hierarchy in these real-world benchmarks would significantly bolster the work.
Ambiguity and Under-specification in Mathematical Formulations:

**W2 Several notational inconsistencies, ambiguities, and lack of detail pervade the key equations.** For example, in Section 3.3, the notation used in Equation 4 ($C^{j}=S_{i}^{1} \circ S_{i+1}^{2} \circ \cdots$) is not sufficiently clarified—specifically, what objects are being concatenated, and along which axes? Similarly, “Repeat” in the representation fusion block is not defined (Page 5, Equation following “Representation Fusion”), potentially confusing the exact operational semantics in high-dimensional embeddings.
The formal description of angle ($\mathfrak{A}$) and distance ($\mathfrak{D}$) in hyperbolic space (Section 3.1) might cause confusion: the role of temperature/scaling parameters, referencing of “root,” and practical computation are under-specified.
Hyperparameters such as $\alpha$, $\beta$, and threshold $t$ in the loss functions (Equations 8–9) are not justified. The reader is left in the dark as to how they are chosen, how sensitive results are to their values, or whether these values generalize. This point is underscored by the limited sensitivity analysis on these hyperbolic-specific hyperparameters, in contrast to standard neural setup (Figure 5).

**W3 Insufficient evidence proves that the method is really useful:** This paper only evaluates on a limited number of traditional benchmarks. It is recommended to test on more benchmarks to effectively demonstrate the model's performance, such as GIFT-EVAL, FEV-Bench, and TFB. Additionally, it lacks comparisons with the latest strong baselines, such as TimeMixer++, OLinear, TimeBridge, and TimePro.

**Questions:**

**Q1** Can the authors quantitatively or visually demonstrate that the benchmark datasets used possess (latent or explicit) hierarchical dependencies suitable for hyperbolic modeling? Are there dataset characteristics or pre-analyses (e.g., hierarchical tree representations) justifying this geometric assumption?

**Q2** What strategies are used to set and tune the hyperbolic-specific hyperparameters $\alpha$, $\beta$, threshold $t$, or curvature $c$? How sensitive is performance to these choices on different benchmarks?

**Q3** Are there any numerical instabilities observed when training with Lorentz-based operations, and what practical measures, if any, are taken to prevent divergence/overflow?

**Q4** Can authors provide concrete empirical metrics—such as wallclock time, GPU memory use, or scalability with increasing $L$ or $C$—to substantiate claimed linear efficiency improvements over quadratic or logarithmic baselines?

**Q5** Would the method degrade gracefully if applied to a setting lacking any inherent hierarchical relationships? For instance, what are the empirical results if applied to univariate forecasting or quasi-flat data?

**Q6** How does HyperTime perform if critical components (e.g., patch representation, MLP in TDM/MDM, or explicit inter-channel dependency modules) are replaced by alternative neural network structures, such as convolutional or transformer mechanisms?

---

> ### Comment · Reviewer_YUyP · 2025-11-22
>
> I recommend that the authors should consider conducting experiments on other benchmarks such as M4, PEMS, or TFB. These additional experiments should have lower GPU requirements than Traffic or ELC, and adding them can provide more substantial evidence for the effectiveness of HyperTime. Many recent advanced end-to-end works add these benchmarks, so that i think it is a standard criterion in the end of 2025. If more experiments are added, I will further increase my rates.

---

> ### Author Response · Authors · 2025-11-22
> **Response to Reviewer YUyP (1)**
>
> We thank Reviewer YUyP for the open-minded and positive engagement with our work. The suggestions provided are invaluable, and we have been actively working on the experiments you mentioned since your last review. We are committed to completing these as quickly as possible and will present the results in our next revision.

---

> > ### Comment · Reviewer_YUyP · 2025-11-22
> >
> > That's my duty. I think your work is interesting and the results on TSLib seem largely outperform previous SOTA TimeMixer++ and OLinear. So that if more experiments are conducted, the performance of HyperTime will be more credible. Good luck!

---

> > > ### Author Response · Authors · 2025-11-23
> > > **Response to Reviewer YUyP (2)**
> > >
> > > Please see our general response for the changes.

---

### Official Review · Reviewer_d2tD · 2025-10-31

**Soundness:** 2
**Presentation:** 3
**Contribution:** 2
**Rating:** 4
**Confidence:** 4

**Summary:**

This paper proposes the HyperTime model for multivariate time series forecasting tasks, which encodes temporal blocks, individual channels, and multi-channel sequences into a unified hyperbolic representation space. By imposing hyperbolic hierarchy and entailment constraints on the encoded representations, it ensures sufficient interaction between point-level information, intra-channel information, and inter-channel information, thereby addressing the issue that existing methods struggle to fully capture the hierarchical relationships among these three aspects while improving efficiency.

**Strengths:**

1. The work innovatively introduces hyperbolic space constraints and effectively divides time series into three hierarchical levels—temporal blocks, individual channels, and multi-channel sequences—while ensuring interaction among these information types through two constraints.
2. The experiments are comprehensive and robustly support most of the claims made in the paper.
3. The workflow of this paper is very clear.

**Weaknesses:**

1. In the HEL loss, a threshold t is set uniformly across the three perspectives (point, intra-channel, inter-channel). However, the referenced MERU method calculates thresholds separately. Could you clarify how t was determined? Would separate thresholds yield better performance?
2. The paper uses S (blocks), R (sequences), and D (multivariate sequences) to model entailment. However, R inherently contains multivariate information, suggesting it may not represent a univariate sequence. Does this imply that the relationship between R and D is not purely entailment, but rather that D augments R with additional multi-channel dependencies captured by MDM?
3. The font size in Table 2 is excessively large.
4. The paper does not specify the configuration of hyperparameters α and β in HTL and HEL, nor does it explore the impact of different hyperparameter settings on model performance.
5. Most operations in the framework are the existing techniques in time series domain. It is better to clarify how novelty lies in the proposed method.

**Questions:**

As in Weaknesses

---

### Official Review · Reviewer_hUJm · 2025-11-03

**Soundness:** 2
**Presentation:** 2
**Contribution:** 2
**Rating:** 2
**Confidence:** 4

**Summary:**

This paper presents **HyperTime**, a framework that leverages **hyperbolic representation learning** for multivariate time series forecasting. The proposed approach aims to model the hierarchical relationships among **time patches**, **single channel**, and **multiple channels** within time series data. By mapping representations from different levels into hyperbolic space, the method seeks to better preserve the intrinsic hierarchical relationship that is difficult to capture in conventional Euclidean geometry. Extensive experiments demonstrate that HyperTime achieves state-of-the-art forecasting performance across multiple benchmark datasets.

**Strengths:**

- The proposed framework leverages hierarchical modeling to capture temporal relationships, providing a promising direction for representation learning in time-series data.

- The experiments are conducted on multiple widely used benchmarks, achieving low computational complexity while showing consistent performance improvements.

**Weaknesses:**

* **Unclear rationale and contributions:** The technical contributions remain vague. It is unclear how the proposed method differs from existing hyperbolic representation learning approaches or what specific advantages it offers. In particular, during the Euclidean encoder’s feature extraction, is the hierarchy between patch-level and channel-level representations already captured?

* **Unsupported key claims:** The experimental evidence does not convincingly support the claimed hierarchical structure. For instance, Figure 4 only visualizes pairwise distances rather than the hierarchy itself, while Figure 2 shows low-level patch features near the center and high-level multi-channel features near the boundary—but the rationale behind this spatial arrangement is not clearly articulated.

* **Insufficient parameter justification:** The selection of the negative curvature parameter ($c$) and the angular threshold ($t$) in hyperbolic space lacks explanation. How sensitive is the prediction performance to these parameter choices?

* **Notation inconsistencies:** The notation appears inconsistent. For example, the symbol $C$ is used with different meanings in Eq. (4) and Eq. (6), which could confuse readers.

**Questions:**

* In Figure 1, the relationship between the output of the HSHC module and the decoding output remains unclear.
* In Equation (9), the angles intended to represent the entailment relationship require further detailed explanation to verify their correctness.

---

### Author Response · Authors · 2025-11-19
**Response to reviewers**

We sincerely thank the reviewers for their insightful feedback.

The reviewers commended HyperTime's "principled motivation" (YUyP) in using hyperbolic geometry for hierarchical time series, recognizing it as a "novel perspective" (QF5e) and "promising direction" (hUJm). The HTL and HEL (YUyP) were highlighted as novel constraints. Our "comprehensive" (YUyP) experiments were praised for achieving "state-of-the-art" (hUJm) results with "linear" (YUyP) efficiency and a "clear workflow" (d2tD) for capturing multi-level dependencies.

We appreciate the positive reception and will address the constructive critiques below. (**Appendix uploaded in supplementary material.**)

* **Motivation**

1. Hierarchy in Time Series (YUyP's W1/Q1)

We've added Fig. 1 (Appx.) visualizing three-level hierarchy (patch, channel, multi-channel) using an ETTh1 sample. This "part-forms-whole" structure is a natural assumption for all benchmarks.

2. Rationale for Hyperbolic (hUJm's W1)

Hyperbolic space is uniquely suited for tree-like structures with minimal distortion. Unlike Euclidean space, which expands polynomially and forces a trade-off between crowding and distance, hyperbolic space expands exponentially. This allows child nodes to be placed near their parent while remaining well-separated, preserving hierarchical fidelity.

* **Novelty** (hUJm's W1, d2tD's W5)

We are the first to apply hyperbolic learning to TSF. Our core innovation is the HSHC module, which explicitly constraints three-level hierarchy. We introduce HEL to learn the "part-forms-whole" relationship and a novel HTL to learn three-level hierarchy. HyperTime performs an "Euclidean extraction + Hyperbolic organization" paradigm differs from and consistent gains over strong Euclidean baselines.

* **Methodology**

(hUJm's Q1) We apologize for the ambiguity. The final loss is the sum of standard autoregression and HSHC module's loss. We will revise the figure to illustrate this clearly.

(hUJm's Q2) Entailment is represented by placing the "whole" vector inside the cone of its "part" vector.

(d2tD's W2) The R represents a single-channel sequence, while D is the multivariate representation.

(YUyP's W2) In Eq. 4, patch representations are concatenated along the length axis (Fig. 1(b)). Due to their larger scale, the channel and multi-channel representations need to be repeated. Each patch is concatenated with its corresponding channel, and each channel with its corresponding multi-channel representation (Fig. 1(d)).

* **Experiments**

1. Full Results and Discrepancies (QF5e's W3)

We have added Table 3 (Appx.) with results across all horizons. Baseline discrepancies stem from our commitment to fair, unified setup using a single default learning rate per dataset (from TSLib). While fine-tuned rates can reproduce original results, it benefits all models, including our own.

2. Experiment (YUyP's W3, Q5)

The 8 datasets are widely-used in TSF, ensuring direct comparison with existing work. We have added OLinear and TimePro, as their focus is more comparable to our model than other 2. The updated results (Table 1, Appx.) confirm HyperTime’s advantage. Furthermore, our model’s consistent performance across datasets with varying hierarchical complexities demonstrates its robustness for weaker hierarchies.

* **Analysis**

1. Rationale for Spatial Arrangement (hUJm's W2)

Our hierarchy flows from generic to specific: patches (near center), channels (intermediate), multi-channels (near boundary). This arrangement, consistent with other hyperbolic methods, is evidenced by distance distribution in Fig. 4.

2. Hyperbolic Parameter Sensitivity (hUJm's W3, d2tD's W1/W4, YUyP's W2/Q2, QF5e's W1/W2)

We generate the angle threshold using MERU’s half_aperture method (in our code). Our sensitivity analysis (Fig. 2, Appx.) shows: larger curvatures benefit complex data; large thresholds hurt performance; HTL and HEL require a balanced combination. The HTL component ratio, like the HEL ratio, has a negligible impact; thus, we present only HEL results. Note that the 0 curvature in Fig. 2(a) represents an Euclidean model with constraints.

3. Training (YUyP's Q3)

To ensure numerical stability with Lorentz operations, we increased the epsilon (eps) value from 1e-8 (MERU) to 1e-5.

4. Ablation Study (YUyP's Q6)

An analysis  (Table 2, Appx.) shows that complex methods (Transformers) are better for large-scale data, while simpler methods suffice for small ones.

* **Presentation** (hUJm's W4, d2tD's W3, YUyP's W2, QF5e's W4)

We will correct notational inconsistencies, standardize citations, and adjust font sizes to improve readability.

* **Closing**

We sincerely thank the reviewers, and will thoroughly revise the manuscript. We hope our responses have adequately addressed your concerns. Should any questions remain, we would welcome the opportunity to discuss them further. If we have successfully resolved your doubts, we would be deeply grateful for your reconsideration of this work.

---

> ### Comment · Reviewer_YUyP · 2025-11-20
>
> Thanks for the feedback, i have read the rebuttal and appendix, and I am willing to make an initial adjustment to the score.
>
> In my opinion, by the end of 2025, the studies on end-to-end supervised forecasting models have hit a bottleneck, meaning they have not achieved substantial improvements. While research in Hyperbolic Space may sound somewhat novel, it seems that some basic components of deep learning (e.g. softmax, layernorm, activation) inherently determine that it is essentially learning correlations on a Lie group structure. In other words, the potential process of deep learning has already been based on some high-dimensional latent geometric structures. We hope the authors can be aware of this and explain its rationality from more theoretical perspectives. If not, more experiments are needed to demonstrate that the performance is at least the strongest. A study on end-to-end forecasting in the end of 2025 should refer to the qualities of OLinear (NIPS 2025) and TimeMixer++ (ICLR 2025), providing more theoretical and empirical evidence.
>
> If more of my opinions can be resolved more properly (W3), I am willing to make further adjustments to the score.

---

> ### Author Response · Authors · 2025-11-21
> **Response to Reviewer YUyP (1)**
>
> Thank you for the constructive feedback and for adjusting the score. We deeply appreciate your openness and have worked diligently to address your concerns.
>
> * **More Experiments**
>
> To address empirical concerns, we have conducted new experiments for TimeMixer++ (We rent a server to facilitate the experiments on ECL and Traffic). The results (Table 1, Appx.) show our model’s superior or competitive performance against TimeMixer++, OLinear, and TimePro. All results will be integrated into the main text, and the related papers will be cited.
>
> * **Theoretical Rationale for Hyperbolic Learning**
>
> We agree that deep learning implicitly discovers latent geometric structures, and will differentiate our approach from implicit learning in two main ways:
>
> 1. Explicit vs. Implicit Geometry: Standard models learn geometry as an emergent property of Euclidean-based optimization. In contrast, our method operates directly within hyperbolic space, providing an inductive bias tailored for the data's inherent structure from the outset.
>
> 2. Matching Geometry to Three-Level Hierarchy: Our choice of hyperbolic geometry is directly motivated by our proposed three-level hierarchy (patch, channel, multi-channel). This hierarchy is inherently designed to capture nested dependencies: From local patch-level to channel-wise and finally global multi-channel relationships. Hyperbolic space, with its ability to efficiently embed such hierarchical structures with minimal distortion, is the natural and theoretically justified geometric space for this architecture. By explicitly embedding our hierarchy in a space with constant negative curvature, we provide a more suitable foundation than a model that implicitly learns in an Euclidean-derived geometry.
>
> We hope the response could address your concerns. If not, please let us know, and we will be happy to make further improvements.

---

> > ### Comment · Reviewer_YUyP · 2025-11-21
> >
> > I appreciate your efforts, but OLinear and TimeMixer++ involve evaluations on more short-term forecasting scenarios, such as PEMS and M4. I hope you can supplement these experiments because there is still plenty of time for the rebuttal phase. This process can effectively enhance the solidity of your paper, not limited to this submission to ICLR 2026.

---

> ### Author Response · Authors · 2025-11-23
> **Response to Reviewer YUyP (2)**
>
> We sincerely thank reviewer YUyP for this insightful suggestion. We agree that evaluating on the M4 dataset could provide a broader perspective. However, our primary focus is on multivariate time series with a three-level hierarchy, which is different from the univariate nature of M4. To ensure a relevant evaluation that aligns with our core contribution, we selected four PEMS subsets (PEMS03, PEMS04, PEMS07, and PEMS08), which are multivariate and exhibit hierarchical characteristics.
>
> Following the reviewer's suggestion, we have conducted additional experiments comparing HyperTime against the latest strong baselines on these four PEMS subsets. Since the TSLib does not provide the PEMS subsets, we downloaded and processed them ourselves. For our experiments, we forecast the traffic flow and split the data into training, validation, and test sets with a ratio of 6:2:2. **The results, now uploaded to the Appendix (Supplementary Material)**, demonstrate that HyperTime achieves performance that is either on par with or superior to these baselines.
>
> We are grateful for the reviewer's valuable feedback, which has undoubtedly strengthened our work. In the revised manuscript, we will incorporate these new results into the main body to better highlight our contributions, enhance the robustness of our evaluation, and increase the overall credibility of our findings.

---

> > ### Comment · Reviewer_YUyP · 2025-11-24
> >
> > Thanks for your detailed feedback. I want to clarify that the suggestion of adding M4 is about my original concerns (W3, Q5). Although you still don't handle this point, I'm basically satisfied after i read the response and appendix, so that i decide to further raise the scores to 6. Furthermore, i think some minor issues can be focused to better improve the quality of your paper, i.e., the aesthetic level of tables (1-3) and pictures (4-6).
> >
> > Overall, I remain open-minded. If the quality of the draft is further improved, I may consider raising the score further.

---

> > > ### Author Response · Authors · 2025-11-24
> > > **Response to Reviewer YUyP (3)**
> > >
> > > Thank you for your thoughtful follow-up and for raising your score. We are very grateful for your open-mindedness and constructive suggestions.
> > >
> > > We fully acknowledge your point regarding the M4 dataset. **We will adjust our model to a two-level hierarchy and conduct the suggested experiments. The new results will be added to the Appendix as soon as possible.** To preserve the paper's focus, we intend for these results to remain in the Appendix.
> > >
> > > We also agree on the importance of presentation and will improve the aesthetic quality of the tables and figures in the final version.
> > >
> > > We are sincerely grateful for your help. Your open and constructive approach is a model for us, both as authors and as reviewers for ICLR 2026. We have learned a great deal from your diligence and will strive to emulate this valuable spirit in our own reviewing work.

---

> ### Author Response · Authors · 2025-11-24
> **Response to Reviewer YUyP (4)**
>
> To address the concerns, we have adapted HyperTime for M4 dataset and conducted new experiments.
>
> * **Model Adaptation (Two-Level Hierarchy):**
>
> 1. Scope: We extracted features only at the patch and channel levels.
> 2. HEL Modification: We retained only the Angle(SR) component.
> 3. HTL Modification: We adjusted the constraint to Distance(S, O) < Distance(R, O), where O is the hyperbolic origin (root) from Fig. 4 of the paper (We will correct the related notation inconsistency).
>
> * **New Experiments & Results:**
>
> We evaluated the adapted model against strong baselines on the M4 dataset. As shown in Table 4 (Appx.), our two-level HyperTime could outperform all baselines, which provides strong evidence for the effectiveness of the patch-channel hierarchy constraint.
>
> We deeply appreciate the reviewer's time and effort in helping us improve our paper. We will also improve the presentation later.

---

### Author Response · Authors · 2025-11-30
**To the Area Chair**

Dear Area Chair,

We were shocked by the information leak incident and deeply appreciate the extra time and effort you have had to dedicate as a result. We hope this concise summary helps to streamline your final assessment.

**1. Nov 12-19**

From the first day of the rebuttal, we have been proactive and fully engaged in addressing the reviewers' concerns. Following the conference's official guidance, we completed a comprehensive response and added extensive new experiments (**Appendix uploaded in supplementary material.**) before the first-week deadline (Nov 20).

**2. Nov 19-24**

During the subsequent discussion period (Nov 19-24), we fully resolved all concerns of Reviewer YUyP. As a result, **Reviewer YUyP raised the score from 2 to 6 and indicated a willingness to increase it further**.

**3. Nov 24-27**

We were focused on improving the paper's presentation and awaiting other reviewers' responses. However, the response we were waiting for never came, instead, we were met with the outbreak of the incident on the 27th.

**4. Since Nov 27**

We have continued to work on improving the paper's presentation. **Other 3 reviewers had not yet replied before the incident. However, we are confident that our detailed response submitted on Nov 19 addressed their major concerns.** We believe they would be as satisfied as Reviewer YUyP upon reviewing our thorough rebuttal.

**5. Closing**

**We would like to add a brief clarification.** The 0 curvature in Fig. 2(a) (Appx.) represents an Euclidean model with constraints. This experimental result directly addresses the core experiment concern of Reviewer QF5e (highlighted in W2). Our findings demonstrate that as the curvature increases, the model's performance improves, especially on complex data. Conversely, the Euclidean model with 0 curvature, is at a disadvantage in handling complex multi-level hierarchical relationships. This provides strong evidence for the rationale of our motivation to introduce hyperbolic learning into the hierarchical modeling of time series.

Thank you for your consideration and leadership during this challenging period.

---

### Meta-Review · Area_Chair_U7hi · 2026-01-02

**Summary:**

The reviews are predominantly negative, specifically from 2, 4, 2,6 with the most expressing dissatisfaction about the weakness of the proposed method. Consequently, hUJm express critical conerns about Unclear rationale and contributions, unsupported key claims and Insufficient parameter justification;
d2tD concerns about the hyper-parameter choice; YUyP express concerns bout limited hyperbolic representation justification and insufficient evidence.
Although the authors have provided detailed reponses to the concerns, only YUyP has expressed the willings to raise the score, while others’ concerns have been partly solved.
Therefore, I tend to recommend the rejection of this paper in its current form.

**Reviewer Concerns:**

Most concerns of YUyP have been solved by raising the score, while others have not been well solved in current format.

**Reviewer Scores:**

YUyP may raise the score.

---

### Decision · Program_Chairs · 2026-01-26

Reject